# Exposure to Dental Filling Materials and the Risk of Dementia: A Population-Based Nested Case Control Study in Taiwan

**DOI:** 10.3390/ijerph16183283

**Published:** 2019-09-06

**Authors:** Natalia Mikhailichenko, Kimitoshi Yagami, Jeng-Yuan Chiou, Jing-Yang Huang, Yu-Hsun Wang, James Cheng-Chung Wei, Te-Jen Lai

**Affiliations:** 1Institute of Medicine, Chung Shan Medical University, Taichung 40201, Taiwan; 2NEVRON International Medical Center, 690078 Vladivostok, Russia; 3Graduate School of Oral Medicine, Department of Oral Health Promotion, Matsumoto Dental University, 1780, Gobara, Hirooka, Shiojiri, Nagano 399-0781, Japan; 4School of Health Policy and Management, Chung Shan Medical University, Taichung 40201, Taiwan; 5Department of Medical Research, Chung Shan Medical University Hospital, Taichung 40201, Taiwan (J.Y.H.) (Y.H.W.); 6Department of Medicine, Chung Shan Medical University Hospital, Taichung 40201, Taiwan; 7Graduate Institute of Integrated Medicine, Chung Shan Medical University, Hospital, Taichung 40201, Taiwan; 8Department of Psychiatry, Chung Shan Medical University Hospital, Taichung 40201, Taiwan

**Keywords:** dementia, amalgam, dental fillings, advanced age, nested case control study

## Abstract

When studying the range of toxic substances triggering dementia, special attention should be paid to the materials used in dental practice, particularly to dental fillings containing amalgam. This necessitated conducting large-scale epidemiologic studies. The aim of our research was to determine the risk factors for developing dementia when filling materials containing amalgam are used in dental practice. In order to achieve the set goals, the following tasks were undertaken: (1) The social and demographic characteristics of the examined patients were studied; (2) the spectrum of concomitant somatic diseases was determined in patients of different gender and age; and (3) the relationship between dementia incidence and the volume of dental filling material containing amalgam was identified in patients with different somatic diseases. In general, the research conducted did not reveal any direct relationship between the development of dementia and the volume of filling material containing amalgam. However, among the people with dementia, there were persons for whom its progression was accelerated in cases where a large volume of dental filling material containing amalgam was present.

## 1. Introduction

By the beginning of the 21st century, dementia had begun to spread epidemically and became a major health care problem in all countries and continents. Dementia is a chronic decline in cognitive function that causes impairment relative to a person’s previous level of social and occupational functioning. There are over 100 forms of dementia. The most well-known form is Alzheimer’s disease, which accounts for 50%–60% of all cases. Other forms include vascular dementia, dementia with Lewy bodies, and fronto-temporal dementia (15%–20%). The disease affects 50 million people worldwide, with a new case occurring somewhere in the world every three seconds. According to forecasts, by 2050, 130 million patients with dementia are expected in the world. It is believed to result in economic costs of US$604 billion a year. The cause of cognitive disorders is unknown in many cases. Toxic–metabolic encephalopathies result from diverse insults that can affect cognitive function. Chronic conditions that alter or damage nerve cells and synapses involved in cognition are the biological basis of dementia. One of the toxic substances affecting the brain tissue is mercury.

According to the World Health Organization, mercury, or hydragyrum (which means “liquid silver”), is a heavy metal that negatively affects the environment and living beings. Mercuric chloride (II) is highly toxic, but it was commonly used in medicine as an antiseptic and antimicrobial agent for the treatment of sexually transmitted diseases in earlier times. Unfortunately, medical doctors of that time could not evaluate the long-term effects of mercury toxicity to human health because they did not have such techniques and methods.

Nowadays in our daily life there are many sources of mercury: Dental fillings with amalgam, vaccines containing thiomersal, contaminated seafood, mercuric-containing bulbs, thermometers, and other sources.

Dental amalgam fillings have been widely used in dental treatments such as cavity restorations, endodontic retrograde root fillings, and core build-up. Mercury’s use in dentistry has been considered controversial since the 19th century. Despite the availability of other reliable materials, amalgam is still used due to its cost efficiency and ease of use.

Dental amalgam is a two-component system consisting of a liquid mercury and metal alloy mixture. The ratio of components is approximately equal: 43%–54% mercury and 57%–46% metal powder. Most properties of amalgam depend on the composition of the mixture. Traditionally, amalgam based on Ag3Sn alloy consists of silver (65%–75%), tin (23%–28%), copper (2%–8%), and other trace metals (zinc, lead).

In 2009, the US Food and Drug Administration (FDA) concluded that dental amalgam is safe in restorative treatment. This conclusion is shared by the American Dental Association (ADA), which claims that amalgam is a safe and valuable material for dental restorations. However, the use of dental amalgam is still being questioned and challenged based on recent epidemiological findings.

Primate research demonstrated a permanent low-level release of mercury from amalgam fillings [1], and the calculation of its release rate was based on single- and multiple-face amalgam restorations [2]. The brain and kidney mercury content determined at autopsy was associated with the number of amalgam surfaces [3]. The detrimental role of amalgam fillings (about 50% mercury) in the development of nervous system disorders has been well documented by several studies [4,5].

Mercury vapor, being highly volatile and lipid soluble, can cross the blood–brain barrier and the lipid cell membranes and can be accumulated in the cells in its inorganic forms.

A recent study using a longitudinal health insurance database showed that women exposed to amalgam restoration were more likely to have Alzheimer’s disease [6]. These toxic effects of HgCl_2_ can be triggered by the inhibition of neuronal outgrowth and the induction of cortical neuron degeneration [7]. Moreover, an exposure to mercury increases the risk of hypertension, myocardial infarction, coronary dysfunction, and atherosclerosis [8].

Mercury ions binding the sulfhydryl groups of membrane proteins and enzymes block oxidative processes, decrease the RNA concentration in cells, and damage protein synthesis in different phases. It was found that apoptosis is the result of oxidative stress and the accumulation of reactive oxygen species (ROS)—hydroxide radicals (HO), superoxide (O_2_^−^), and hydrogen peroxide (H_2_O_2_). Commonly, oxidative stress induced by Hg takes place in mitochondria. Mercury penetrates into cells, accumulates in the mitochondria, and causes cell membrane damage, uncoupling of the electron transport chain, accumulation of ROS, oxidative stress, activation of apoptotic caspases, and, as a result, cell death [9,10].

Various studies have reported the association of amalgam use with autoimmune and neurological diseases, such as Alzheimer’s disease [11], Parkinson’s disease [12], and chronic fatigue syndrome [13]. However, only few population-based studies of dental amalgam effects on neurodegenerative diseases are available. This study aimed to investigate the association between dental amalgam use and dementia in a Taiwanese population.

## 2. Materials and Methods

### 2.1. Database and Settings

This nested case control study used reimbursement data obtained from the Longitudinal Health Insurance Database 2000 (LHID 2000) spanning the period 1997–2013. LHID2000 is a subset of the National Health Insurance (NHI) Research Database (NHIRD) managed by the Taiwanese National Health Research Institutions. This dataset contains all information concerning socio-demographic status; outpatient, inpatient, and emergency care; surgical treatment; and prescription drugs. As many as 1 million beneficiaries were randomly selected from the NHIRD registry (about 23 million beneficiaries) in 2000. The International Classification of Diseases, Ninth Revision, Clinical Modification (ICD-9-CM) codes were used to identify the disease diagnoses. A peer review system improves the reliability of diagnosis coding in this dataset. All insurance claims were monitored by medical reimbursement specialists. Our study was approved by the Institutional Review Board of Chung Shan Medical University Hospital (CSMU, IRB No. 17114).

### 2.2. Study Population

Patients diagnosed with new-onset dementia (ICD-9-CM: 331.0, 290.0–290.4) between January 1997 and December 2013 were categorized as the case group (*n* = 20,262). In order to improve the validity, the diagnosis of dementia was defined according to at least two outpatient visits or a single admission. The date of the first visit for dementia was defined as the index date. Patients with dementia before January 2003 were excluded (*n* = 3541) because the information on exposure conditions that we collected was over less than 5 years, so it was too short to explore the association between dental restoration and dementia. Finally, there were 16,666 dementia cases and twice as many age–sex-matched non-dementia controls (*n* = 33,332) included in this study. Figure 1 illustrates our study design.

### 2.3. Definition of Dental Disease and Dental Filling Materials

We defined dental caries, pulpitis (ICD-9-CM: 521.0, 522.0, 522.1), and gingival and periodontal diseases (ICD-9-CM: 523) as dental disorders or conditions. The dental filling materials included those used for amalgam restoration (claim codes: ‘89001’, ‘89002’, ‘89003’, ‘89101’, ‘89102’, ‘89103’) and resin restoration (claim codes: ‘89004’, ‘89005’, ‘89008’, ‘89009’, ‘89010’, ‘89012’, ‘89014’, ‘89015’, ‘89104’, ‘89105’, ‘89108’, ‘89109’, ‘89110’, ‘89112’, ‘89113’, ‘89114’, ‘89115’) before the index date. The number of dental restoration procedures was considered to be the major factor in this study. We also considered dental extraction (claim codes: ‘92013’, ‘92014’, ‘92092’) as the oral hygiene status.

The potential confounding factors considered included demographic variables (i.e., age at index date, sex, place of birth, and income), co-morbidities, and healthcare utilization (i.e., frequency of outpatient visits within 5 years, length of hospital stays (days) within 5 years, frequency of dental visits within 5 years) before the index date.

### 2.4. Statistical Analysis

Categorical variables were expressed as numbers and percentages and compared using the chi-squared or Fisher’s exact test. Conditional logistic regression was used to estimate crude and adjusted odds ratios (ORs) with a 95% confidence interval (CI) for the case group compared with the control group. In multivariate analysis, we adjusted for co-variates that may have predisposed a patient to dementia. In order to deal with the residual confounding, the inverse propensity score weighting was used in this study, and the standardized difference value was determined to evaluate the difference between the case and control groups. Cohen (1988) suggested that effect size indices of 0.2 can be used to represent small effect sizes (Table 1).

Statistical analysis was performed using SAS (version 9.4; SAS Institute, Cary, NC, USA). A *p* value of <0.05 indicates statistical significance.

## 3. Results

This study examined the data on 49,998 patients of dental clinics who received treatment during the period from January 1997 to December 2013. The patients studied were split into two groups: The first group (main) included the patients diagnosed with dementia (*n* = 16,666), while the second (control) group included the patients who did not develop dementia within the same period of observation and treatment (*n* = 33,332). The overall workflow indicating how the cases and controls were drawn from the population database (with the exclusion criteria used for the case and control groups) is shown in Figure 1.

The case (dementia) and control groups were compared across a range of parameters which could be potential confounding factors in the development of dementia, including demographic variables, healthcare utilization, and different comorbidities (Table 1 and Table 2).

Senior citizens predominated among the subjects in both groups (93.63% of subjects were aged >60 and 43.11% of subjects were aged >80). Subjects of both sexes were equally represented. The urban population predominated among the study subjects of both groups (52.15%–53.16% urban dwellers vs. 15.39%–16.29% rural population). The percentage of patients with low income was as low as 0.89%–1.13% (Table 1).

The frequency of outpatient visits in the majority of examined patients was >100 (69.67% and 55.63% in the case and control groups, respectively). However, the length of inpatient stays was twice as high in the subjects of the case cohort as compared to the control group subjects (Table 1).

Significant differences in comorbidity frequencies between subjects with dementia and those never diagnosed with dementia were found: Psychiatric disorders (18.98% vs. 35.98%, aOR 1.904), DM (22.81% vs. 32.99%, aOR 1.263), ischemic stroke (15.59% vs. 38.30%, aOR 2.252), CKD (8.46% vs. 13.47%, aOR 1.087), and Parkinson’s disease (2.45% vs. 9.87%, aOR 2.772). These indicators reflect the possibly complex nature of the development of dementia and the heterogeneity of the combination of somatic factors. Other indicators were not convincing and had minimal differences: Thyroid disease (2.54% vs. 3.16%, aOR 1.023), chronic liver disease (10.01% vs. 12.75%, aOR 1.042), asthma (9.29% vs. 11.43%, aOR 0.920), etc. Their role in the development of dementia is not completely clear, and further long-term studies are needed.

As far as the correlation between the development of dementia and the number of amalgam restorations is concerned, the aOR values were 0.980 and 1.019 for the subjects with less than three fillings and with more than four fillings, respectively. For resin restoration, the aOR values were 0.972 and 1.025 for the subjects with less than eight restorations and with more than nine restorations, respectively (Table 2). Both comparisons in amalgam restorations and resin restorations were non-significant according to the results; however, there was a positive association with an increase in the number of fillings.

## 4. Discussion

This study analyzed the effect of the use of amalgam-containing dental filling materials (as well as a number of socio-demographic parameters of the patients, their utilization of healthcare services, and the range of concomitant diseases) on the development of dementia. While our study did not reveal any direct link between the development of dementia and the overall use of amalgam-containing filling material, we observed a negative impact of the extensive use of such filling materials (>3 amalgam restorations and >8 resin restorations) on the development of dementia.

In the course of a large-scale epidemiological study, we established heterogeneous mechanisms of the development of dementia symptoms. The study results are reliable and well grounded due to the representative subject samples, the use of valid research methods fitting the study problems and tasks, and the correct application of contemporary statistical methods.

This study’s findings show that it is important to prevent the side effects of amalgam in its use as a convenient caries treatment material. However, an association was not confirmed from the epidemiological statistics included in this study. For proof, it is necessary to argue about the scientific character of the materials in the amalgam.

High-copper amalgam arose after revision in 1977 from an initial low-copper type. The low-copper type of amalgam comprises γ 1 aspect and γ 2 aspect (Sn8Hg), while the high-copper type comprises γ 1 aspect and η aspect (Cu6Sn5). The classic amalgam was developed by French dentist Onesiphore Taveau in 1826. The development of copper-rich, multi-dispersion, reinforced alloys for amalgam by Innes and Youdelis et al. [14,15,16] provided the corrosion resistance and physical properties of classical amalgam. An amalgam alloy in which palladium and indium were added was subsequently developed, but there were no major changes in the alloy composition.

The alloy particles with the γ 2 aspect of silver–Hg of conventional amalgams are soft, and they are less corrosion resistant. However, Hg and the hardening reaction with the alloy are persistent and need several hours before Hg disappears in response to the alloy. Moreover, the isolation of the elution of mercury or mercury vapor is different between restorations because the surface of the setting of an amalgam body is not a uniform conformation. Therefore, unlike in experiments on mercury vapor exposure, it becomes difficult to determine the relationship between quantity and influence. Many researchers have reported that mercury vapor is released from dental amalgam. Vimy et al. reported [17] that the oral mercury vapor levels of patients with amalgam filling were higher than those of controls without amalgam.

Besides this, the intraoral mercury levels increase with further chewing, and the development of mercury vapor from amalgam decreases by discontinuing chewing. Particularly, the abundance of the mercury vapor increases further by chewing gum, more when the patient chews common food [18]. The mercury vapor which occurs in a buccal capsule increases the uptake of mercury to the body. Moreover, this mercury vapor dissolves in saliva and is absorbed by deglutition into the body [19]. Frykholm [20] found a significant increase of urine and fecal Hg levels after restoration with mercurial amalgam in humans and animals using a mercury radioisotope tracer.

On the other hand, the mercurial level in the blood of persons with amalgam fillings has been reported to be high, and a correlation was associated with the amalgam filling number and surface area [21,22]. As compared with controls without amalgam filling, in pathologic autopsy after the death of persons with amalgam fillings, Hg levels in the occiput cortex were approximately two times higher, and those in the kidney were approximately 10 times higher [3]. Eggleston and Nylander [23] reported that the mercurial levels in the gray matter and white matter of the brain in persons with amalgam fillings were higher than those in controls. Interestingly, the number of amalgam restorations and the surface area of the amalgam were found to correlate with the mercury amounts in the brain and the kidney.

Further, the mercurial intake from amalgam is higher than the Hg quantity taken in from food, water, and air and is a main source of Hg exposure in public environments [24]. Placental transmission of mercury vapor has been shown to occur, raising concerns about the effect of amalgam restoration on the growth and development of fetuses and infants [25,26]. Among allergic occurrences relating to amalgam fillings with mercury, stomatitis and anal eczema were reported first [27].

The Hg levels in the breath [28], blood [21], urine [29], and brain tissue [23] of persons with amalgam restoration were shown to be significantly higher than those in persons who do not have amalgam restoration. The Hg elution mechanism from amalgam restoration is completely unknown, but contributions by aspect metamorphoses (e.g., Ag_2_Hg_3_(γ_1_) → AgHg(β)) and corrosion (M_1_Hg_m_ → 1M^n+^ + mHg^o^ + 1ne^−^) have been suggested. Even if Hg is released in either mechanism, the Hg is isolated in a metal state and elutes it [30]. A part of the Hg which is eluted vaporizes and is absorbed through the respiratory organs in the body. The Hg which remains behind in saliva is swallowed and is absorbed through digestive organs. The diseases in organisms caused by Hg eluted from amalgam are extremely few, but inflammation and metallic taste, hemorrhaging, hypersalivation, and gingivitis in gingiva and buccal mucosa due to amalgam fillings have been reported [31,32,33,34].

Sun et al. [6] evaluated the association between dental amalgam fillings and Alzheimer’s disease (AD) in a large-scale cross-sectional study of the Taiwanese population aged 65 years and older and made a conclusion that women’s exposure to amalgam was significantly associated with AD. Their findings in this first population-based study are significant, since a previously reported retrospective New Zealand cohort study could not find an association between dental amalgam exposure and AD due to an insufficient number of cases [14]. In comparison with previous studies that indicated higher mercury concentrations in serum and cerebrospinal fluid samples from living subjects and post-mortem brain samples of patients with Alzheimer’s disease, Sun et al. performed an exclusive assessment of epidemiologic variables with the use of the Longitudinal Health Insurance Database. While they analyzed multiple-face amalgam restorations based on the different NHI codes, they did not discriminate between single and multiple fillings and, thus, analysis of the difference between single- and multiple-surface amalgam restoration is, unfortunately, unavailable. Despite their awareness of the limitations of their epidemiological study (described in their paper), their findings suggest that further investigations will be needed to more accurately investigate the association between AD and amalgam use via assessment of the duration of amalgam exposure, the number of dental fillings, the size of restorations, the surface area of the restored lesion, and other possible risk factors involved in AD pathophysiology.

Clear conclusions based on scientific data have not been obtained, but the benefits and potential risks of amalgam fillings are being discussed all over the world. There have been many negative reports about the health effects of exposure to mercury from dental amalgam. However, negative effects may only appear in persons with high sensitivity to Hg when a larger population is exposed to low concentrations of Hg.

Limitations of this research. This investigation was based on concordances of treatment history and dementia (including Alzheimer’s disease) in an insurance database. Therefore, this study did not include an element for the diagnosis category of the critical cause of dementia. The critical causes of dementia, including Alzheimer’s, are various and include foreign poisons such as mercurial ions, neurodegenerative affections, vascular disorders, and genetic brain function disorders. We cannot confirm in this analysis whether Hg or some other inducement in the etiology of dementia occurrence was released by amalgam. Therefore, evaluation of instances of newly filled amalgam and changes in blood Hg levels for related verification with dementia will be necessary for elucidation.

In our future work, we aim to carry out additional follow-up studies in order to elucidate the mechanisms underlying the relationships between oral cavity diseases, the number of fillings (including those made with amalgam-containing materials), and lost teeth and the development of cognitive dysfunctions.

## 5. Conclusions

The conducted research did not reveal any direct link between the development of dementia and the volume of amalgam-containing filling material. However, an acceleration of disease progression was observed in patients with dementia for whom extensive use of amalgam-containing dental filling materials was reported. The potential for amalgam to provoke dementia in genetically predisposed subjects was suggested by the high incidence of concomitant somatic diseases indicative of the development of neurodegeneration (psychiatric disorders, extrapyramidal system pathology—Parkinson’s disease, and cardiovascular diseases).

## Figures and Tables

**Figure 1 ijerph-16-03283-f001:**
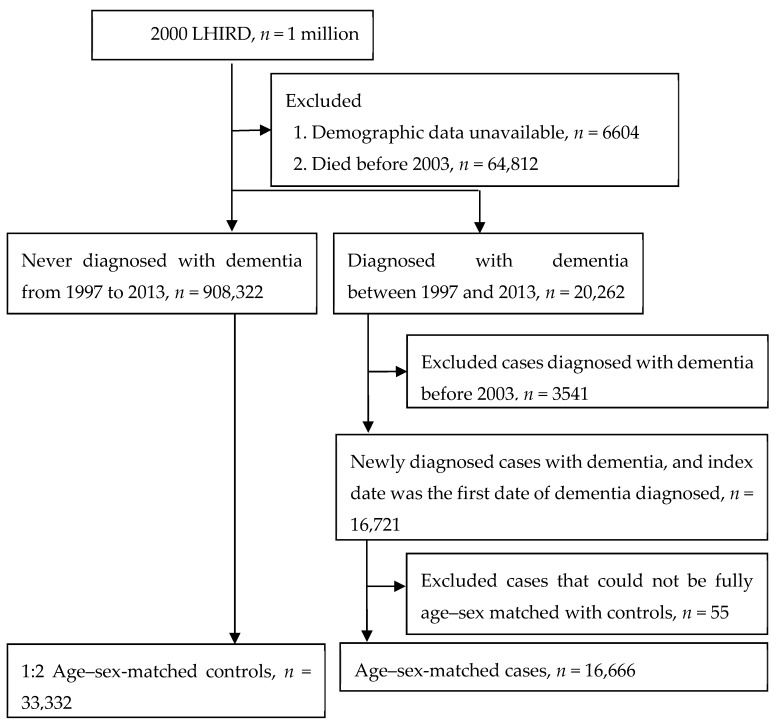
Study design.

**Table 1 ijerph-16-03283-t001:** Characteristics among groups.

	Group		Standardized Difference	
	Control	Dementia Cases	Original	After Propensity Score Weighting
Age at index date			0.0000	0.00949
<40	236 (0.71%)	118 (0.71%)		
40–59	1888 (5.66%)	944 (5.66%)		
60–79	16,840 (50.52%)	8420 (50.52%)		
≥80	14,368 (43.11%)	7184 (43.11%)		
Sex			0.0000	−0.00657
Female	17,674 (53.02%)	8837 (53.02%)		
Male	15,658 (46.98%)	7829 (46.98%)		
Urbanization			0.02665	0.00216
Urban	17,719 (53.16%)	8692 (52.15%)		
Suburban	10,484 (31.45%)	5259 (31.56%)		
Rural	5129 (15.39%)	2715 (16.29%)		
Low income	296 (0.89%)	189 (1.13%)	0.02459	0.00255
Outpatient visits			0.38812	0.03348
0	1475 (4.43%)	22 (0.13%)		
1–27	2935 (8.81%)	882 (5.29%)		
25–54	3503 (10.51%)	1258 (7.55%)		
55–98	6875 (20.63%)	2893 (17.36%)		
≥99	18,544 (55.63%)	11,611 (69.67%)		
Length of hospital stay			0.53386	0.01054
0	18,243 (54.73%)	5310 (31.86%)		
1–6	5237 (15.71%)	2655 (15.93%)		
7–13	3808 (11.42%)	2401 (14.41%)		
≥14	6044 (18.13%)	6300 (37.8%)		
Comorbidities				
Urticaria	3296 (9.89%)	1830 (10.98%)	0.03573	0.00326
Rheumatic diseases	1514 (4.54%)	979 (5.87%)	0.05998	0.00312
Thyroid disorders	846 (2.54%)	526 (3.16%)	0.03717	−0.00123
Inflammatory diseases	11020 (33.06%)	5609 (33.66%)	0.0126	0.00253
*Helicobacter pylori* infection	129 (0.39%)	122 (0.73%)	0.04627	0.0023
Peptic ulcer	6677 (20.03%)	4658 (27.95%)	0.18621	0.00916
Hepatitis B virus infection	494 (1.48%)	318 (1.91%)	0.03301	0.00125
Hepatitis C virus infection	552 (1.66%)	389 (2.33%)	0.0485	0.00169
Psychiatric disorders	6325 (18.98%)	5996 (35.98%)	0.38796	0.00261
Asthma	3098 (9.29%)	1905 (11.43%)	0.07013	0.00057
Chronic liver disease	3336 (10.01%)	2125 (12.75%)	0.08643	0.00338
Diabetes mellitus	7603 (22.81%)	5498 (32.99%)	0.22844	0.0052
Hyperlipidemia	7812 (23.44%)	4509 (27.06%)	0.08336	0.00565
Heart failure	2731 (8.19%)	2141 (12.85%)	0.1521	0.00631
Hypertension	18,708 (56.13%)	11,559 (69.36%)	0.27624	0.01299
Coronary artery disease	7832 (23.5%)	5140 (30.84%)	0.16567	0.01067
COPD	7047 (21.14%)	4828 (28.97%)	0.18137	0.00566
Ischemic stroke	5197 (15.59%)	6383 (38.30%)	0.52944	−0.00166
Cancer	2784 (8.35%)	1568 (9.41%)	0.03713	0.00871
CKD	2820 (8.46%)	2245 (13.47%)	0.16087	−0.00041
Parkinson’s disease	816 (2.45%)	1645 (9.87%)	0.31248	−0.02095
Dental visits			0.07134	0.00614
0	12,714 (38.14%)	5902 (35.41%)		
1–4	8343 (25.03%)	4500 (27.00%)		
5–9	5337 (16.01%)	2725 (16.35%)		
10–14	3092 (9.28%)	1434 (8.60%)		
≥15	3846 (11.54%)	2105 (12.63%)		
Dental caries, pulpitis	9007 (27.02%)	4421 (26.53%)	−0.01118	0.00233
Gingival and periodontal diseases	11,964 (35.89%)	6070 (36.42%)	0.01099	0.00281
Amalgam restoration			0.0138	0.01226
0	23,651 (70.96%)	11,750 (70.5%)		
1–3	6743 (20.23%)	3384 (20.3%)		
≥4	2938 (8.81%)	1532 (9.19%)		
Resin restoration			0.02049	0.02005
0	17,588 (52.77%)	8659 (51.96%)		
1–8	10,257 (30.77%)	5147 (30.88%)		
≥9	5487 (16.46%)	2860 (17.16%)		
Extraction			0.05957	0.02361
0	14,456 (43.37%)	6800 (40.80%)		
1	4365 (13.10%)	2125 (12.75%)		
≥2	14,511 (43.53%)	7741 (46.45%)		

**Table 2 ijerph-16-03283-t002:** Logistic regression for estimating the odds ratio of dementia.

	Crude		Conditional Multivariate Modeling		Inverse Propensity Score Weighting Modeling	
	OR *	95% CI **	aOR ***	95% CI	aOR	95% CI
Age at index date (ref: <40)						
40–59	-	-				
60–79	-	-				
≥80	-	-				
Sex (ref: Female)						
Male	-	-				
Urbanization (ref: Urban)						
Suburban	1.023	0.981–1.066	0.975	0.930–1.023		
Rural	1.079	1.023–1.138	1.023	0.962–1.087		
Low income	1.280	1.066–1.538	1.316	1.057–1.639		
Outpatient visits (ref: 0)						
1–27	20.094	13.109–30.801	14.185	9.334–21.557		
25–54	24.013	15.694–36.743	14.071	9.265–21.369		
55–98	28.138	18.439–42.939	13.859	9.145–21.002		
≥99	41.868	27.48–63.789	14.723	9.711–22.320		
Length of hospital stay (ref: 0)						
1–6	1.742	1.647–1.842	1.403	1.320–1.492		
7–13	2.166	2.041–2.299	1.618	1.513–1.730		
≥14	3.581	3.418–3.752	2.272	2.144–2.407		
Comorbidities						
Urticaria	1.124	1.058–1.194	0.967	0.904–1.035		
Rheumatic diseases	1.312	1.208–1.425	1.056	0.962–1.159		
Thyroid disorders	1.251	1.121–1.398	1.023	0.904–1.158		
Inflammatory diseases	1.027	0.987–1.068	0.993	0.915–1.079		
*H. pylori* infection	1.898	1.481–2.433	1.813	1.374–2.392		
Peptic ulcer	1.549	1.483–1.617	1.020	0.969–1.074		
Hepatitis B virus infection	1.293	1.122–1.491	1.009	0.857–1.188		
Hepatitis C virus infection	1.419	1.245–1.618	1.007	0.865–1.174		
Psychiatric disorders	2.399	2.301–2.502	1.904	1.812–1.999		
Asthma	1.259	1.186–1.338	0.920	0.857–0.988		
Chronic liver disease	1.314	1.24–1.392	1.042	0.972–1.118		
Diabetes mellitus	1.666	1.599–1.736	1.263	1.202–1.327		
Hyperlipidemia	1.212	1.161–1.264	0.928	0.881–0.977		
Heart failure	1.652	1.556–1.754	1.033	0.961–1.11		
Hypertension	1.769	1.701–1.84	1.098	1.045–1.154		
Coronary artery disease	1.452	1.393–1.513	0.956	0.909–1.005		
COPD	1.521	1.458–1.587	1.065	1.011–1.122		
Ischemic stroke	3.360	3.219–3.508	2.252	2.142–2.368		
Cancer	1.140	1.068–1.216	0.806	0.749–0.868		
CKD	1.684	1.588–1.787	1.078	1.007–1.155		
Parkinson’s disease	4.364	4.004–4.756	2.772	2.519–3.051		
Dental visits (ref: 0)						
1–4	1.162	1.108–1.219	1.006	0.946–1.07		
5–9	1.100	1.040–1.163	0.975	0.896–1.06		
10–14	0.999	0.932–1.071	0.906	0.816–1.006		
≥15	1.179	1.109–1.254	1.006	0.901–1.123		
Dental caries, pulpitis	0.975	0.935–1.017	0.941	0.882–1.003		
Gingival and periodontal diseases	1.023	0.984–1.064	0.989	0.905–1.080		
Amalgam restoration (ref: 0)						
1–3	1.010	0.964–1.059	0.986	0.931–1.045	0.980	0.947–1.015
≥4	1.050	0.983–1.121	1.006	0.928–1.092	1.019	0.971–1.069
Resin restoration (ref: 0)						
1–8	1.019	0.977–1.063	0.984	0.928–1.043	0.972	0.941–1.004
≥9	1.059	1.005–1.116	1.029	0.949–1.116	1.025	0.984–1.068
Extraction (ref: 0)						
1	1.035	0.975–1.098	0.970	0.904–1.040	1.014	0.973–1.056
≥2	1.134	1.090–1.180	1.021	0.966–1.078	1.056	1.024–1.088

* OR, odds ratio; ** CI, confidence interval; *** aOR, adjusted odds ratio.

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
