# Peer review of "Exposure to Dental Filling Materials and the Risk of Dementia: A Population-Based Nested Case Control Study in Taiwan"

_ijerph, 2019, doi:10.3390/ijerph16183283_

Round 1

Reviewer 1 Report

Revisions: AD need 40 years to develop (Braak et all, see Mutter et al
in many publications). Therfore, not only 5 years of time period before
Ad diagnosis is important, but also longer time periods (Amalgam in this
time). Also, Amalgam can be under crowns and brakes and even in jaw
bone. Thus, "Amalgam free" persons may have still amalgam exposure.
This should be discussed.

Does Inorganic Mercury Play a Role in Alzheimer’s Disease? A Systematic Review and an Integrated Molecular Mechanism. (Journal of Alzheimer’s Disease 22 (2010) 357–374); DOI 10.3233/JAD2010100705

Alzheimer Disease: Mercury as pathogenetic factor and apolipoprotein E as a moderator. (Neuroendocrinology Letters No.5 October Vol.25, 2004)

Comments on the Article “The Toxicology of Mercury and Its Chemical Compounds” by Clarkson and Magos (2006). (Critical Reviews in Toxicology, 37:537–549, 2007); DOI: 10.1080/10408440701385770

Author Response

Dear Reviewer,

we would like to thank you for your time, your review report and useful articles. We hope it will help us to improve our paper.

What about the problems you suggest to discuss we can say that in our article we describe the only one of many factors of dementia (and Alzheimer’s disease as well) development. We understand that AD is slowly progressing process but mercury as a toxic factor can accelerate the disease and also affect patients who are genetically determined to neurodegenerative diseases. It is estimated that the half –life of mercury in the brain can be as long as 20 years [Friberg L, Mottet NK. Accumulation of methylmercury and inorganic mercury in the brain. Biol Trace Elem Res. 1989; 21:201–206].

As for “Amalgam free” persons although there is no experimental report of verification of this theory, since amalgam as a prosthesis of a crown prosthesis is completely covered inside the prosthesis, it is not structurally exposed like an amalgam filling. It is considered appropriate to exclude them as factors.

The English language was edited and improved.

Thank you!

Reviewer 2 Report

A. Exposure of Dental Filling Materials containing amalgam and its impact in the human health Risk has been one of the research hotspots in the field of environmental science and human health. In particular, it is not clear how the risk of Dementia associated with use of amalgam-containing in dental practice. On the whole, this may be a valuable research attempt and worth to be published, but there are still some problems needed to be modified. B. Comments on the manuscript 1. The description of the research method in section abstract is not clear, so it is suggested to rewrite. 2. We suggest that the name of the paper further highlight the research area. As we all known, the risk of Dementia varies from different region (the amount of amalgam in soil is different), climate and lifestyle. In this manuscript, all the data comes from the database LHID 2000 of Taiwan, China. So add research area is extremely necessary. 3. Line 100-105 section 2.3: Please modify the paper layout. The Formats of figure 1, table 1 and references do not meet the requirements of the journal. Such as, the figure name is usually placed figure below.

Author Response

Dear Reviewer,

we wish to express our appreciation to you for your time and your practical comments. We hope it will really help us to improve our paper.

We rewrote the Abstract session, added the research area to the title of the article and replaced the Figure 1 (as well as tables 1,2) to meet the requirements of the journal according to your suggestions.

References were also corrected according to the journal’s requirements.

Thank you!

Reviewer 3 Report

The authors have examined great number of population, patients of dental clinic tested for 16 years.

In my opinion in the publication there is more information from literature than from own investigations (especially in Discussion - lines 180 --270).

A lot of references are older than 20 years, but the authors found also association with new references (no 5, 6, 13 ..) .

Down in the table 2 I suggest give once more an information about abbreviations (OR  aOR,….)

In article autors didn’t discussed and point out wrong statements connected with discussing problems (see the statements in line 253-254)

Author Response

Dear Reviewer,

we wish to express our gratitude to you for your time and your tolerance. Thank you for marking not only problems but also advantages of our paper.

We added footnotes with abbreviation expansion down in the Table 2 according to your comments. We also deleted the “wrong” statement cause this problem is worth to be discussed separately.

Thank you!

Reviewer 4 Report

This MS aims to explore the risk of Dementia associated with the use of amalgam-containing in dental practice. This paper needs to be reviewed in several aspects:

What are the constituents of dental amalgam involved in dementia? what are the levels of Hg? Which is the mechanism of action to induce dementia? What scientific evidence supports these facts? As the authors refer in the "Limitations of the Study" etiology of the dementia occurrence and relationship of an amalgam cannot be confirmed. An update of papers are needed for a more robust introduction, including pertinent references on neurodegenerative diseases and their relationship to mercury-based dental amalgams. In addition dietary habits, environmental exposure to Hg may also impact health. The English language should be extensively improved.

Other comments:

Lines 42-45 – Please rephrase to avoid repetition of word “dementia”.

Introduction

References related to dementia, namelly Alzheimer’s disease are missing in the first paragraph. In addition no papers were cited concernig amalgam fillings.

Please clearly define the study objectives.

Line 116 - to evaluate

Line 224 - Please rephrase: “The case that caused

Please consider the following papers:

ErtaÅŸ E et al. Human brain mercury levels related to exposure to amalgam fillings. Hum Exp Toxicol. (2014)

Gul N et al. Quantification of Hg excretion and distribution in biological samples of mercury-dental-amalgamusers and its correlation with biological variables. Environ Sci Pollut Res Int. (2016)

Jirau-Colón H, González-Parrilla L, Martinez-Jiménez J, Adam W, Jiménez-Velez B. Rethinking the Dental Amalgam Dilemma: An Integrated Toxicological Approach. Int J Environ Res Public Health. 2019 Mar 22;16(6).

Author Response

Dear Reviewer,

we would like to thank you for your time and your careful examination of our paper. Your recommendations were concrete and very helpful for us to improve our manuscript.

Introduction section was extended according to your comments, we added information about constituents of dental amalgam, mechanism of Hg inducing dementia etc.

We also made some minor changes such as rephrasing of repeated words etc. The English language has been edited by the MDPI English editing service and improved.

Thank you for new papers you've recommended for our consideration.

Reviewer 5 Report

The methods of the statistical analysis are good and more informative. The manuscript is well written except typo errors. 

Author Response

Dear Reviewer,

we wish to express our gratitude to you for your time and your friendly review report. We hope our research is worth to be published.

The English language was edited and improved.

Round 2

Reviewer 4 Report

Authors have done several changes and the MS was improved.